# Tribological Properties of Si_3_N_4_-hBN Composite Ceramics Bearing on GCr15 under Seawater Lubrication

**DOI:** 10.3390/ma13030635

**Published:** 2020-01-31

**Authors:** Fang Han, Huaixing Wen, Jianjian Sun, Wei Wang, Yalong Fan, Junhong Jia, Wei Chen

**Affiliations:** College of Mechanical & Electrical Engineering, Shaanxi University of Science & Technology, Xi’an 710021, China; hanfang@sust.edu.cn (F.H.);

**Keywords:** composite ceramic, tribofilm, hydrodynamic lubrication, seawater

## Abstract

This paper concerns a comparative study on the tribological properties of Si_3_N_4_-10 vol% hBN bearing on GCr15 steel under seawater lubrication and dry friction and fresh-water lubrication by using a pin-on-disc tribometer. The results showed that the lower friction coefficient (around 0.03) and wear rate (10^−6^ mm/Nm) of SN10/GCr15 tribopair were obtained under seawater condition. This might be caused by the comprehensive effects of hydrodynamics and boundary lubrication of surface films formed after the tribo-chemical reaction. Despite SN10/GCr15 tribopair having 0.07 friction coefficient in the pure-water environment, the wear mechanismsits were dominated by the adhesive wear and abrasive wear under the dry friction conditions, and delamination, plowing, and plastic deformation occured on the worn surface. The X-ray photoelectron spectroscopy analysis indicated that the products formed after tribo-chemaical reaction were Fe_2_O_3_, SiO_2_, and B_2_O_3_ and small amounts of salts from the seawater, and it was these deposits on the worn surface under seawater lubrication conditions that, served to lubricate and protect the wear surface.

## 1. Introduction

With land resources on the way to depletion, the development of marine-based resources has been gaining increasing attention. The specialized equipment used for marine engineering is the foundation of emergent marine resources and economic development. The equipment used in such environments requires special engineering, in which the friction-pair component is an important aspect [1]. Compared to traditional metal or alloy materials, which are susceptible to electrochemical corrosion, structural ceramics possess unique properties and are promising materials for use in frictional components and corrosion-resistant parts. Si_3_N_4_-based ceramics have found wide applications in a variety of engineering fields due to their properties. They are high in hardness, low in density, excellent in thermal and chemical stability, and outstanding in corrosion resistance, Moreover, these ceramics have seen wide-ranging industrial applications, including use in high-speed cutting tools, engine parts, sealing modules, bearings, and corrosion-resistant components [2,3,4,5]. Wu et al. [6] found a WC-10Co-4Cr/Si_3_N_4_ tribopair with a friction coefficient of 0.09 and a wear rate lower than 9 × 10^−6^ mm^3^N^−1^m^−1^ in natural seawater, attributing this to Si_3_N_4_ tribochemical reaction. Liu et al. [7] conducted research on the tribological behaviors of Si_3_N_4_/AISI316 tribopair in seawater conditions. They found that the friction coefficient under seawater lubrication decreased to 0.16, attributing this to SiO_2_ gel lubrication function. Accordingly, Si_3_N_4_ ceramics have a promising prospect of finding wide applications in marine engineering equipments. In previous studies, researchers just assessed the friction and wear properties of single-phase Si_3_N_4_ ceramics in fresh water condition [8,9,10,11,12], restricting their; their engineering applications due to their drawbacks shown dry friction condition, such as higher friction coefficient and wear rate.

Hexagonal boron nitride (hBN) is generally believed a lubricant. In some cases, adding hBN into the ceramic matrix does have improved the properties of ceramics in friction and wear. Alexandra et al. [13] found that the addition of hBN into Si_3_N_4_-hBN ceramics improved the wear resistance. Li et al. [14] investigated the dry tribological properties of self-mated couples of B_4_C-hBN ceramics in dry condition. They found that, the friction coefficient varied with hBN content: the higher the hBN content was the lower the friction would while the higher wear coefficient. In the case a B_4_C ceramic matrix after adding hBN, the friction coefficient lowered from 0.373 for the B_4_C/B_4_C tribopair to 0.005 for the B_4_C-20 wt% hBN/B_4_C tribopair in water lubricating condition [15]. In our previous research [16,17,18,19,20], we studied the tribological performances of Si_3_N_4_-hBN bearing on stainless steel at different sliding speeds and with a normal load. When the speed was 1.73 m/s and a load of 10N in dry friction conditions, the friction coefficient measured was around 0.12 for the Si_3_N_4_-10 vol% hBN/ASS (austenitic stainless steel) tribopair [16,17]. However, in drip-feed water lubricating condition, the friction coefficient of Si_3_N_4_-20 vol% hBN decreased from 0.35 to 0.01 [18]. Furthermore, with the increase of hBN to 10 vol% at a relative humidity RH of 55~65%, the friction coefficient and wear rate were further reduced respectively to 0.03 and 10^−6^ mm^3^/Nm [19,20]. Evidently, the addition of hBN to the ceramic materials has improved their the tribological properties. Although Si_3_N_4_-hBN ceramics are potential candidates for tribological applications, systematic research detailing the effect of seawater on their tribological behavior is not readily available. H. Tomizawa and T. E. Fischer [21] carried out experiments on the friction of silicon nitride and silicon carbide in water at room temperature, finding that they dissolved during wear tests. A thinner silicon nitride lubricating film formed on the worn surface. Johannes Kurz et al. [22] found, lower wear rate at room temperature with water lubricant. Tribochemical film delamination plus tribochemical reactions were observed as wear mechanisms. Chen Wei et al. [23] found that, after adding hBN to Si_3_N_4_, the friction coefficient significantly decreased, and on the worn surface formed a layer of SiO_2_ and H_3_BO_3_. The film may be caused by the chip inserted in the spalling pit on the sample. The worn pieces underwent reaction with the moisture in air, thus forming a tribochemical film functioning lubrication. They [24,25] also found that, in the marine condition, the tribological properties of silicon nitride ceramics could be enhanced by the second time the addition of hBN. Specifically, the sliding pair of Si_3_N_4_-20wt%hBN/titanium alloy showed the best tribological properties. In deionized water condition, the participation of hBN could also make improvements, but without forming superficial tribo-chemical films. S. Jahanmir et al. [26] found that, in the hydrocarbon condition, silicon nitride interacted with water vapor, producing an amorphous silicon oxide mixed with Si_3_N_4_ crystallites. Obviously, in the formation of silicon carbide, hydrocarbon did not involve in the reaction. In our experiments, the major concern was placed onto the variation the friction and wear properties of Si_3_N_4_ composite ceramics when hBN with varied contents was added in seawater condition at different speeds and loads. It was found that, in seawater condition, minimum values of friction coefficient and wear rate were shown in the Si_3_N_4_-10 vol% hBN/GCr15 tribopair. As such, this study was a comparative study on, the tribological properties of Si_3_N_4_-10 vol % hBN bearing on GCr15 steel in seawater lubrication condition and those in dry friction and fresh water lubrication condition. The analysis was made on the basis of the results of surveying the friction coefficients and wear rates, observing the microstructure, and determining the chemical components of the film formed on the steel. Moreover, a tentative explanation of the formation mechanisms of the film was proposed. Hopefully, the findings of our experiments would trigger wider engineering applications of Si_3_N_4_-based composite ceramics.

## 2. Materials Prepareds and Methods Applied

### 2.1. Materials Prepared and Specimens Made

The detailed synthetization of Si_3_N_4_-hBN composite ceramics is presented somewhere else [16,17,18]. Generally, starting materials were ball milled, hot-press sintered at 30 MPa and 1800 °C, and also were cut into pins. The sintering raw materials were composed of Si_3_N_4_ powder (the content of 90% α-Si_3_N_4_ or more and the average particle size of 0.5 μm) and hBN powder (the purity of 99.6% and the average particle size being of 1 μm). The sintering aids were composed of hBN powder (the purity of 99.6% and the average particle size of 1 μm), Al_2_O_3_ powder (the purity of 99.5%), and Y_2_O_3_ powder (the purity of 99.9%) [27].

However, in this study, Si_3_N_4_-hBN specimens contained only 10 vol% hBN (SN10). Figure 1 shows the results of the sintered speciment Si_3_N_4_-10% hBN ceramic. Its components are β-Si_3_N_4_ and hBN. Table 1 and Table 2 show the physical and mechanical properties of Si_3_N_4_-10% hBN specimens and GCr15. The Si_3_N_4_-10% hBN ceramic was cut into 5 mm × 5 mm × 10 mm rectangular type pins with a surface roughness of lower than 0.08 μm for the friction and wear tests. A GCr15 disc with diameter 44 mm in and thickness of 6 mm was used as the mating material. Its friction part was mechanically ground until surface roughness reached 0.04 μm. Table 3 displays the major components of the disc.

The seawater in the experiments was artificially prepared based on standard ASTM D1141-1998 [29] with pH adjusted to 8.2 by a 0.1 mol/L NaOH solution. Table 4 displays its major components. 

### 2.2. Test Methods

In the experiments, a tribometer was used. Its upper rotary pin contacts a stationary disc in three lubricating conditions: dry, pure water, and seawater. The schematic diagram depicting this testing apparatus is shown in Figure 2. A detailed overview of the apparatus can be found in our previous report [17]. As shown in Figure 2, the SN10 pins are used to slide against the GCr15 disc. When testing the sliding friction, the disc was submerged in the lubrication medium (either fresh water or seawater). All experiments were conducted at ambient temperature with. 10 N set load, the sliding speed of 1.73 m/s, and the grinding distance of 1000 m. Before doing the experiments, all samples were placed in acetone and ethanol and cleaned ultrasonically for 15 min. The tribometer recorded the friction coefficient (*f*), and the wear rate *w* was derived from the formula:*w* = Δ*m*/(*ρFS*),(1)
in which Δ*m* represents the wear volume assessed by according to the weight loss from a microbalance with an accuracy of 0.1 mg, *ρ* is the density, *F* the normal load, and *S* friction distance. In the calculation of *f* and *w*, the initial values were excluded, for the friction coefficients and wear rates were the average of the values from three independent experiments. Phase constitution of sintered Si_3_N_4_-10%hBN ceramic was determined by X-ray diffraction (XRD) (D8 Advance, Bruker, Germany) analysis using Cu Kα radiation. Scanning electron microscopy (SEM) (FEI Apreo, Hillsboro, OR, USA) was adopted to analyze the morphologies of the worn surfaces. X-ray photoelectron spectroscopy (XPS) (AXIS Supra, Manchester, UK) was applied to chemical characterization of the worn surfaces.

## 3. Findings and Discussion

### 3.1. Characteristics of Friction and Wear

Figure 3 shows the friction coefficients of SN10 on GCr15 steel in dry friction, pure water, and seawater conditions, respectively, As revealed in this figure, the friction coefficient of SN10/GCr15 tribopair is greater than 0.5 in dry condition, and higher than that in the lubrication condition. In fresh water lubrication condition, the friction coefficient is approximately 0.07. However, in seawater lubrication condition, the friction coefficient is about 0.03 when the experimental parameters remain unchanged, and its trace is quite stable. Evidently, the seawater environment promoted superior lubrication compared with the pure-water condition.

Figure 4 lists the wear rates of the SN10/GCr15 tribopair in the three lubrication conditions. It can be found that, in the aqueous environments, the tribopairs demonstrated a much better wear resistance compared to the dry condition. The wear rates of the pin or disc were as high as 10^−5^ mm^3^N^−1^m^−1^ under dry friction, while the pins displayed negligible wear rates (not higher than 10^−6^ mm^3^N^−1^m^−1^) in the pure-water and seawater environments. As it is known, most materials exposed to corrosive aqueous environments inevitably suffer corrosion. During the assessment of the sliding friction of the SN10/GCr15 tribopair in the seawater environment, corrosive wear plus mechanical wear affected metal components. However, the lowest wear rate of the pin or disc was obtained in experiments with seawater, as opposed to pure water, indicating that Si_3_N_4_-10% vol hBN possessed an excellent lubricating effect in seawater under the given experimental parameters.

### 3.2. Characterization of the Worn Surfaces with SEM

Figure 5 shows the respective images of the SN10/GCr15 tribopair in dry friction, fresh water, as well as seawater conditions. Obviously, aqueous environments made surfaces relatively more smooth compared with those developed in dry condition, as the aqueous solutions provided lubrication and helped dissipate heat by friction during sliding. Thus, the friction coefficient and wear rate of SN10/GCr15 tribopair were low and limited. Seen from Figure 5b, the worn surface of GCr15 disc was subjected to severe plastic deformation, delamination, and plowing. Likewise, seen from Figure 5a debris found on the surfaces of the SN10 pins, might be traced to the embedding of worn debris in the spalling pits. These results are likely due to the lower shear strength of the GCr15 steel in comparison with the SN10 ceramic, with debris from the formed adhesion spots torn from the GCr15 disc being transferred and adhered to the worn out pins in dry friction condition. In fresh water environment, the worn surfaces of the SN10 pins showed significant wear debris in the transferred layer and slight furrows paralleling the sliding direction (Figure 5c). Furthermore, some discontinuous films were found covering the worn parts of GCr15 disc (Figure 5d). This phenomenon might lie in the formation of worn debris on grinding interfaces, as water in the experiments failed to rid the interfaces of the debris completely, promoting the mechanical abrasion on the interfaces. Figure 5e,f is the relatively smooth worn surfaces of the SN10/GCr15 tribopair in seawater lubrication condition, having only a few scratches and pits. This manifests that the lubricating effect of, seawater is superior to that of fresh water.

In condition, the observed adhesive and abrasive wear dominated the wear mechanisms in dry sliding friction condition, explaining, the high friction coefficient and wear rate. In water lubricating condition, the friction coefficients and wear rates of SN10/GCr15 tribopair were low, since tribofilm was formed on the surface serving to lubricate and protect the worn surface and the hydrodynamic lubrication conferred by the liquid medium. In the sliding friction test, the relative velocity of SN10/GCr15 tribopair produced tribofilms between the contact interfaces, and they in turn, generated lubrication regimes surrounding the boundaries impeding adhesion wear and lowering the friction coefficient. One more phenomenon deserves notice, two regions on SN10 pin were worn out, as shown in Figure 6. Region 1 was primarily composed of Fe, Si, O, and C, indicating that this region had a mixed composition of oxides. Furthermore, the Ca and Cl from seawater could be identified in region 2 as well, indicating that some salt was deposited on or had been incorporated into the wear interface. Fe element was removed while SN10 pin grinding on GCr15 disc. However, in region 1, the Fe element and salt from the seawater were not found, but the oxygen and boron were detected.

### 3.3. XPS Analysis on the Worn Surfaces

Figure 7, Figure 8 and Figure 9 tell the results of the XPS analysis about the worn surfaces of the GCr15 discs grinding against the SN10 pins in lubrication conditions. In dry friction condition, Si 2p, B 1s, and Fe 2p_3/2_ peaks of GCr15 could be subdivided into. On spectra of Si 2p, the peaks were at 98.6 and 102.4 eV, corresponding to Si_3_N_4_ and SiO_2_, respectively. The peaks of B_1s_ were at 190.3 and 192.4 eV, corresponding to BN and H_3_BO_3_, respectively. The peaks of Fe 2p_3/2_ were at 707.2 and 710 eV, corresponding to Fe and Fe_2_O_3_, respectively. In the aqueous environments, the peaks of Si 2p could be attributed to Si, Si_3_N_4_ and SiO_2_. The peaks of B_1s_ corresponded to BN and B_2_O_3_. The peaks of Fe 2p_3/2_ at 707.2 and 710 eV could be assigned to metallic Fe, FeO and Fe_2_O_3_. In our previous report [1], the Si_3_N_4_ and BN underwent following reactions with water:
Si_3_N_4_ + 6H_2_O → 3SiO_2_ + 4NH_3_,(2)
SiO_2_ + nH_2_O → SiO_2_·nH_2_O,(3)
2BN + 3H_2_O → B_2_O_3_ + 2NH_3_,(4)
B_2_O_3_ + 3H_2_O → 2H_3_BO_3_.(5)

The negative values of Gibbs free energy in chemical reactions mentioned above, revealed possible reaction of Si_3_N_4_ and BN with moisture in the air or water in generating SiO_2_, B_2_O_3_, and H_3_BO_3_ [30,31,32]. According to our previous study [20], spalling pits were first to have appeared formed on SN10 pins, since hBN spalled first, and the worn debris were then embedded in the pits to react with water during the grinding friction process. Thus, a triboflim made up of Fe_2_O_3_, SiO_2_, and B_2_O_3_ began to take shape on the worn surfaces, endowing the sliding pair with hydrodynamic lubrication. In summary, in the fresh water and seawater conditions, the low friction coefficient and wear rate of the SN10/GCr15 tribopair were triggered by triboflims developed on the worn surfaces.

### 3.4. Characterization of Seawater Lubrication

In this test, CaCO_3_ and Mg(OH)_2_ were found to be aggregated and enriched on the friction surface with salt ions contained seawater and they took the effect of lubrication at the friction interfaces [33]. Figure 10 shows the XPS spectrum of Ca 2p and Mg 2p in the seawater condition. The peaks of Ca 2p were at 347.6 eV and 350.7 eV, corresponding to those of CaCO_3_, and the peaks of Mg 2p were at 87.9 eV and 88.3 eV, corresponding to those of Mg(OH)_2_, indicating that some amount of salt from the seawater was deposited on or incorporated into the contact interface [34]. XPS results supported the interpretation of the EDS analysis (see Figure 6) by revealing that CaCO_3_ and Mg(OH)_2_ were in deed generated. Following are the related chemical reactions: Mg^2+^ + 2H_2_O → Mg(OH)_2_ + 2H^+^,(6)
HCO_3_^−^ → CO_3_^2−^ + H^+^,(7)
Ca^2+^ + CO_3_^2−^ → CaCO_3_.(8)

Figure 11 shows the highly magnified SEM image of the worn surface of GCr15 steel after grinding in seawater. There was clear turtle-shell sludge deposited on GCr15 steel after bearing on the SN10 pin. Deposits CaCO_3_ and Mg(OH)_2_ have reported to play an important role in the process of tribopair friction and wear in seawater [24]. The layer of deposits CaCO_3_ and Mg(OH)_2_ served to segregate GCr15 disc from seawater and prevent chlorine ions from sprawling over the disc, thus impeding the corrosion rate of GCr15 disc. This might help to explain why the friction coefficient and wear rate of the SN10/GCr15 tribopair were lower in seawater and higher in fresh water.

In addition, it had been reported that the SiO_2_ formed via the tribochemical reaction could be accumulated and aggregated into a SiO_2_ gel deposited on the grinding interface when either SiC or Si_3_N_4_ was involved in grinding in aqueous condition [7,35]. The SiO_2_ gel proper could be an excellent lubricant, so friction and wear were lowed. Ions such as mainly Na^+^ and Cl^−^ in seawater could accelerate SiO_2_ colloids’ aggregation on the friction surface. Then SiO_2_ colloids would turn into SiO_2_ gel lubricant. Although colloidal SiO_2_ could develop in fresh water, it could be easily removed by the water surrounding it, so SiO_2_ gel deposited would be too thin to function as an effective lubricant for the interface. XPS results presented in Figure 8 and Figure 9 indicate that the SiO_2_ generated after the tribochemical reaction could continue staying to function as a good boundary lubricant, contributing to the low friction coefficient and wear rate observed in the present research. To further verify the hypothesis that SiO_2_ colloids would aggregated and turn into SiO_2_ gel in the seawater, a SiO_2_ colloid solution was prepared by mixing tetraethoxysilane with ethanol and water. This solution was then added to the fresh water and seawater. As it turned out that the fresh water was still clear while. The seawater turned milky white. This phenomenon was similar to the results reported by Liu [7]. However, a better understanding of the mechanism must be further explored. The schematic diagrams representing the wear models of the SN10 ceramic bearing on the GCr15 steel under seawater lubricating condition are depicted in Figure 12. As shown, due to the relatively poor combination of 10 vol% hBN with the Si_3_N_4_ matrix, the hBN was easily extruded from the friction surface of the SN10 ceramic, resulting in the formation of spalling pits being formed on SN10 pin. In the course of grinding friction experiments, worn pieces were found either embedded into the pits on SN10 pin or deposited on GCr15 disc (see Figure 5c,e), and subsequently reacted with the available water. The subsequent chemical products aggregated in the pits and spread over the surface (see Figure 5d,f), and eventually they turned into a tribofilm made up of SiO_2_ and H_3_BO_3_. This helps to explain why the friction coefficient and wear rate of SN10/GCr15 tribopair were low in an aqueous condition. As a result, this tribofilm protected the wear interface from abrasive wear and functioned as a lubricant to worn surfaces of the pin and disc. In addition, salt ions from the seawater also accelerated CaCO_3_ and Mg(OH)_2_ accumulation on the friction surfaces, functioning as a lubricant for the boundaries. It in turn effectively overcame corrosion of the chlorine ions. The results of the XPS analysis showed that, in different lubrication conditions, the tribochemical products developed on SN10/GCr15 tribopair were also different. That is to say, the wear mechanisms of the various tribochemical products were also different. Table 5 shows that, in dry friction condition, despite some tribochemical products on the worn surface, there were also ample amounts of worn pieces deposited (see Figure 5a,b), and a complete surface film was not formed, indicating that mechanical wear (adhesive and abrasive wear) was the dominant wear mechanism under these conditions. In an aqueous environment, the film generated by the tribochemical reaction effectively protected the worn surface from abrasive wear, due to its contribution to the hydrodynamic lubrication and boundary lubrication.

## 4. Conclusions

The present paper concert a study on the tribological properties of Si_3_N_4_-10 vol% hBN grinding on GCr15 in dry friction, fresh water, and artificial seawater condition. The following are the summaries of this research: In dry friction condition, the friction coefficient and wear rate of SN10/GCr15 tribopair are higher than that in aqueous condition. The friction coefficient is around 0.5, and wear rate over 10^−5^ mm^3^/Nm. Mechanical wear (adhesive and abrasive wear) dominates wear mechanism in dry friction condition. Moreover, the worn surface are liable to grave plastic deformation, delamination, and plowing. 

In an aqueous environment, when SN10 grinding on GCr15, the values of friction coefficients and wear rates are low that those in dry lubrication condition. Their respective values are *f* ≈ 0.06, *w* ≈10^−6^ mm^3^/Nm in fresh water lubrication condition and, *f* ≈ 0.03, *w* ≈ 10^−6^ mm^3^/Nm in seawater lubrication condition. The film generated by the tribochemical reaction can overcome further abrasive wear to the worn surface, and functions as an effective lubricant. The lubrication characteristics shown in the seawater conditions lies in tribofilm formed in situ at the wear interface or in the layer made up of CaCO_3_ and Mg(OH)_2_ deposited on the worn surface.

## Figures and Tables

**Figure 1 materials-13-00635-f001:**
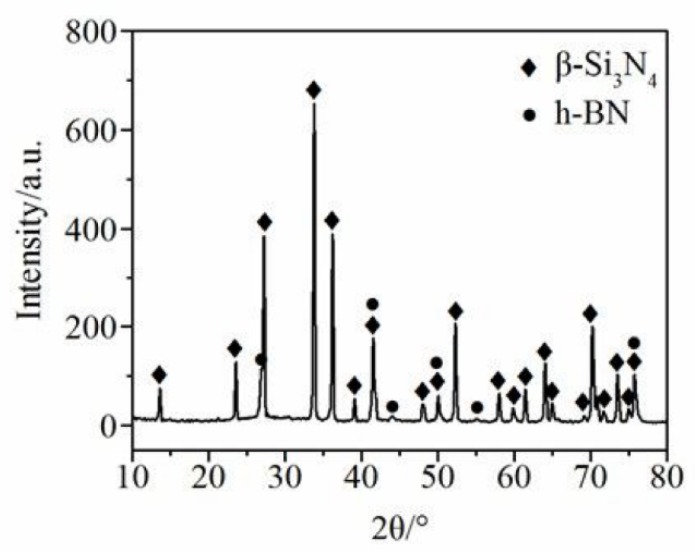
XRD analysis result of the SN10 ceramic.

**Figure 2 materials-13-00635-f002:**
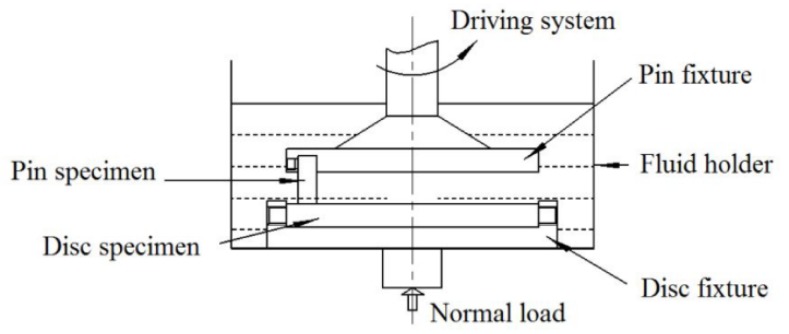
Schematic diagram of the friction and wear test apparatus.

**Figure 3 materials-13-00635-f003:**
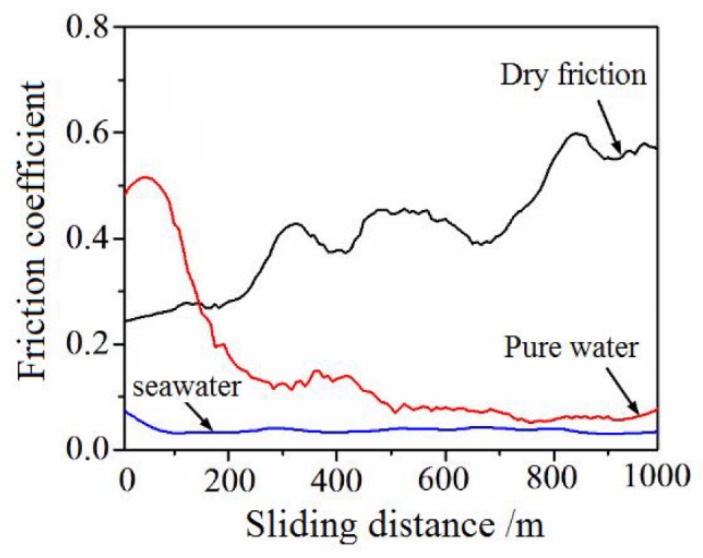
Friction coefficients of the SN10/GCr15 tribopair under the different conditions.

**Figure 4 materials-13-00635-f004:**
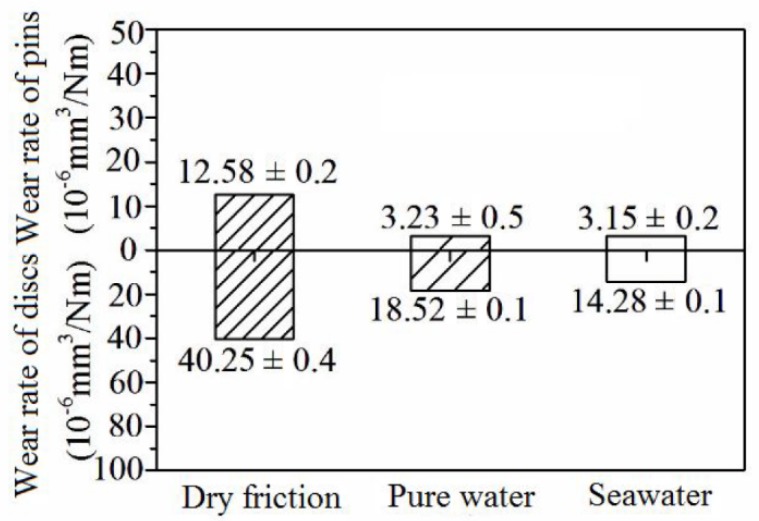
Wear rates of the SN10/GCr15 tribopair under different lubrication conditions.

**Figure 5 materials-13-00635-f005:**
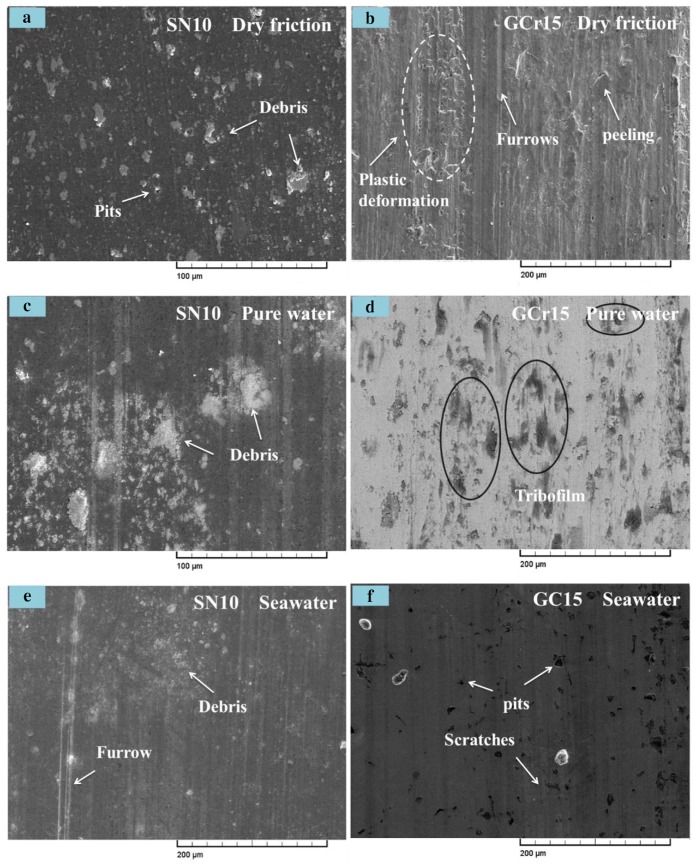
Scanning Electron Microscopy (SEM) images of the worn surfaces of SN10 (**a**,**c**,**e**) and GCr15 (**b**,**d**,**f**) under dry friction (**a**,**b**), pure-water lubrication (**c**,**d**), and seawater lubrication (**e**,**f**).

**Figure 6 materials-13-00635-f006:**
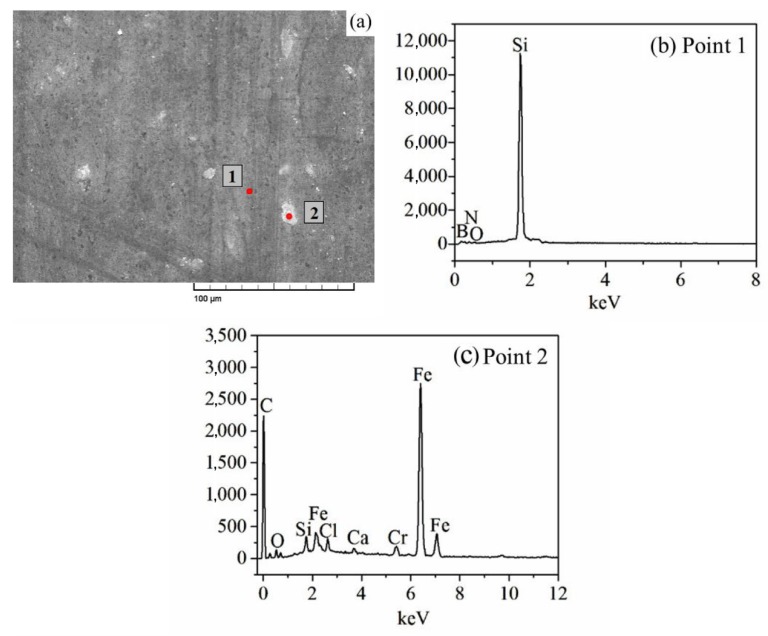
SEM image and EDS (Energy Dispersive Spectroscopy) results of the worn surface of the SN10 pin with seawater lubrication. SEM image of the worn surface of the SN10 pin with seawater lubrication (**a**), EDS results of the worn surface of the SN10 pin with seawater lubrication (**b**,**c**).

**Figure 7 materials-13-00635-f007:**
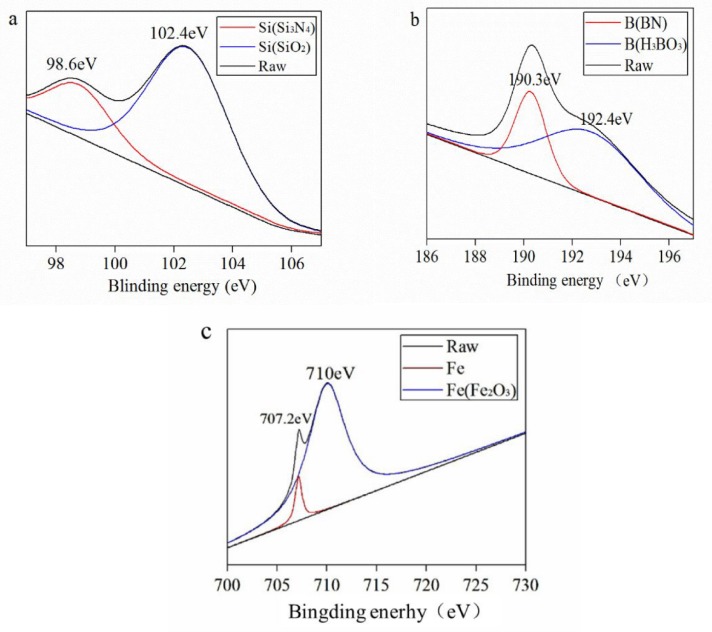
X-ray photoelectron spectroscopy (XPS) spectrum of Si 2p (**a**), B 1s (**b**), and Fe 2p_3/2_ (**c**) on the worn surface of the GCr15 disc sliding against the SN10 pin under dry friction conditions.

**Figure 8 materials-13-00635-f008:**
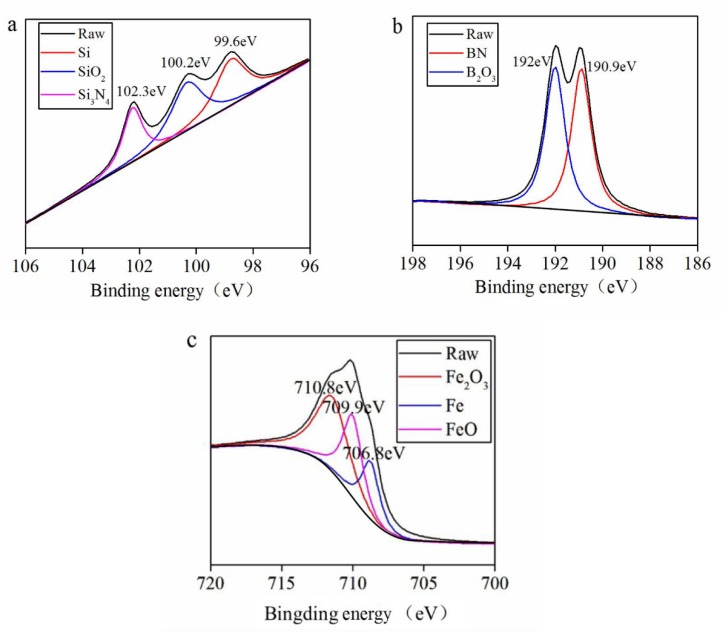
XPS spectrum of Si 2p (**a**), B 1s (**b**), and Fe 2p_3/2_ (**c**) on the worn surface of the GCr15 disc sliding against the SN10 pin in the pure-water environment.

**Figure 9 materials-13-00635-f009:**
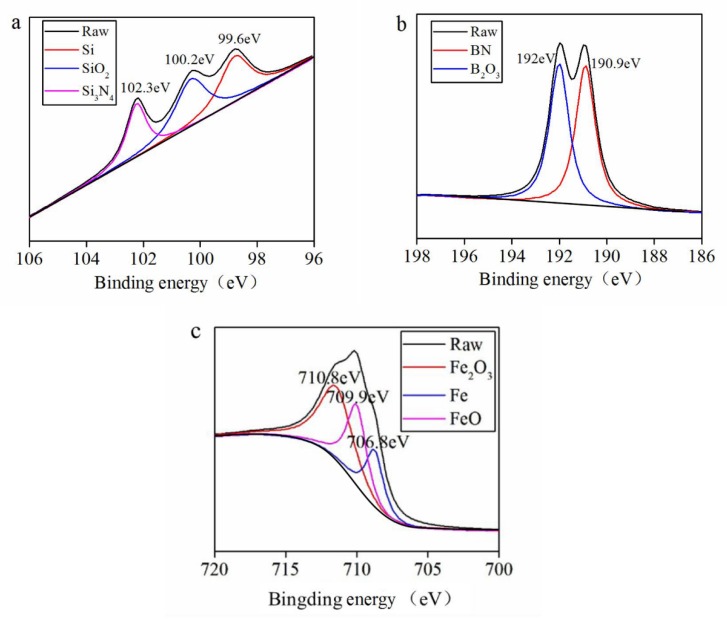
XPS spectrum of Si 2p (**a**), B 1s (**b**), and Fe 2p_3/2_ (**c**) on the worn surface of the GCr15 disc sliding against the SN10 pin in the seawater environment.

**Figure 10 materials-13-00635-f010:**
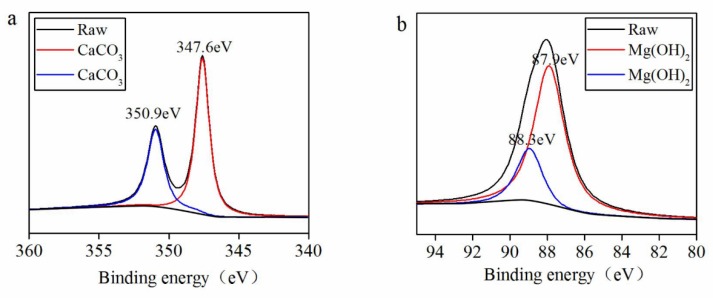
XPS spectrum of Ca 2p (**a**) and Mg 2p (**b**) on the worn surface of the GCr15 disc sliding against the SN10 pin in a seawater environment.

**Figure 11 materials-13-00635-f011:**
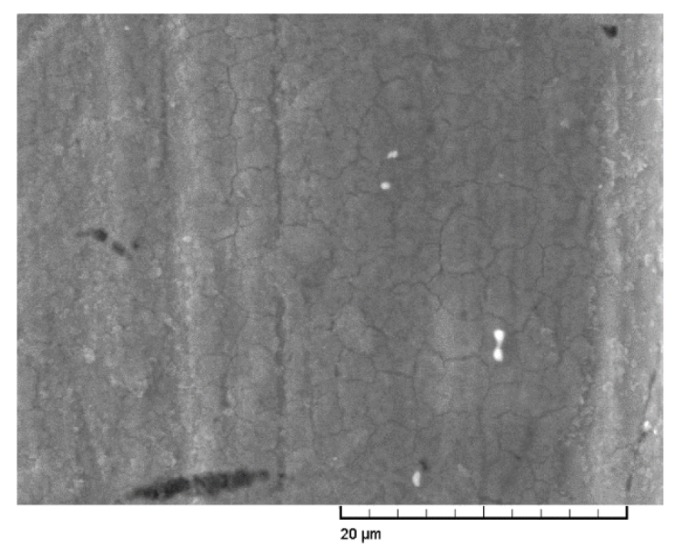
SEM image of the worn surface of the GCr15 at high magnification in a seawater environment.

**Figure 12 materials-13-00635-f012:**
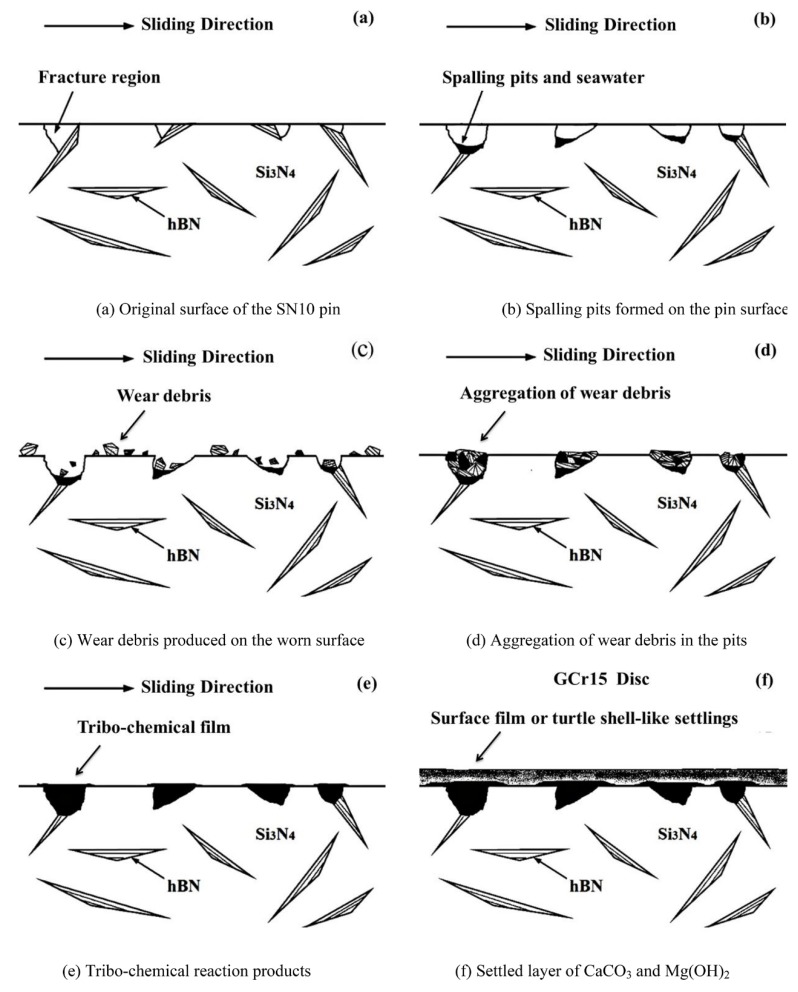
Schematic diagrams depicting wear models of the SN10/GCr15 tribopair in a seawater environment. Original surface of the SN10 pin (**a**), Spalling pits formed on the pin surface (**b**), Wear debris produced on the worn surface (**c**), Aggregation of wear debris in the pits (**d**), Tribo-chemical reaction products (**e**), Settled layer of CaCO_3_ and Mg(OH)_2_ (**f**).

**Table 1 materials-13-00635-t001:** Physical and mechanical properties of the SN10 composite.

Specimens	Density (g/cm^3^)	Porosity (%)	Bending Strength (MPa)	Vickers Hardness (GPa)	Fracture Toughness (MPa·m^1/2^)
SN10	3.10	0.91	613	15.3	7.14

**Table 2 materials-13-00635-t002:** Physical and mechanical properties of the GCr15 steel [28].

Specimens	Density (g/cm^3^)	Tensile Strength (MPa)	Bending Strength (MPa)	Rockwell Hardness (GPa)	Impacting Energy (J)
GCr15	7.81	1902	1617	63	26

**Table 3 materials-13-00635-t003:** Chemical composition of the GCr15 steel.

Grade	Components (wt%)						
	C	S	Si	P	Mn	Ni	Cr
GCr15	0.95	≤0.020	0.27	≤0.027	0.36	≤0.30	1.49

**Table 4 materials-13-00635-t004:** Chemical composition of the artificial seawater.

Constituent	Concentration (g/L)
NaCl	24.53
MgCl_2_	5.20
Na_2_SO_4_	4.09
CaCl_2_	1.16
KCl	0.695
NaHCO_3_	0.201
KBr	0.101
H_3_BO_3_	0.027
SrCl_2_	0.025
NaF	0.003

**Table 5 materials-13-00635-t005:** Tribo-chemical products and wear mechanisms of the SN10/GCr15 tribopair under different lubrication conditions.

Tribopair	Conditions	Tribo-Chemical Products	Wear Mechanisms
SN10/GCr15	seawater	SiO_2_, B_2_O_3_, Fe_2_O_3_, FeO, CaCO_3_, Mg(OH)_2_	hydrodynamic lubrication, boundary lubrication
SN10/GCr15	dry friction	SiO_2_, H_3_BO_3_, Fe_2_O_3_	adhesive wear, abrasive wear
SN10/GCr15	pure water	SiO_2_, B_2_O_3_, Fe_2_O_3_, FeO	hydrodynamic lubrication

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
