# Peer review of "Tribological Properties of Si3N4-hBN Composite Ceramics Bearing on GCr15 under Seawater Lubrication"

_materials, 2020, doi:10.3390/ma13030635_

Round 1

Reviewer 1 Report

In this manuscript, Chen and his co-workers present the investigation of the tribological properties of a Si3N4-hBN composite ceramics sliding against GCr15 steel under dry friction, pure water, and seawater conditions. This manuscript is a continuation of their previous investigation series of the Si3N4-hBN ceramics, and less detailed than previous works. Properties and synthesis of the Si3N4-hBN ceramics have already provided in previous works. The novelty of this work is the application of commercial GCr15 steel in the tribopair. I suggest acceptance of this work, but only after major revision which takes into account comments below and corrects the English of the text.

Comments:

--- line 12: correct GC15 to GCr15

--- Authors should provide the word ‘steel’ for GCr15 at its first appearance, like GCr15 steel, in line 89 and in the abstract, or in the title.

--- Authors should check and correct throughout the manuscript and provide the composition symbol consistently as vol% (if this is the case). % means mass%. It is confusing that e.g. line 88 indicates vol% and line 93 indicates mass%. This latter is improperly used throughout the text.

--- lines 41 and 42: correct formulas by placing numbers to subscript

--- I agree with the sentence in line 98 quoting previous articles describing experimental details; however, the synthetic method has to be provided briefly in this manuscript too. E.g. the third sentence in this paragraph could be: Briefly, starting materials were ball milled, hot-press sintered at 30 MPa and 1800 °C, and cut into pins.

--- Source and grain size of starting materials have to be provided in the Materials section. Please provide information if alpha or beta Si3N4 was used as the starting material. It also has to be commented that vol% composition throughout the paper indicates the vol% of hBN in the initial Si3N4/hBN mixture.

--- line 105: source of the GCr15 disc has to be provided

--- line 113, Table 1: reference is missing

--- line 114, Table 2: reference is missing

--- line 115, Table 3: reference is missing

--- lines 136-139: provide information about the XRD, PES, and SEM instruments (instrument type and resolution).

--- Figures 5 and 6: remove machine generated text from SEM images and provide a proper scale (e.g. using free software as ImageJ)

--- Figure 7 contains axes with increasing binding energy, but Figures 8 and 9 with decreasing energy. Correct them to have the same style.

--- line 212: 3/2 is in subscript in Fe 2p3/2. Check text thoroughly.

--- line 248: check and correct Mg 2s to Mg 2p

--- Table 5 and text: It is difficult to accept that H3BO3 forms under dry conditions from BN, but B2O3 forms under wet conditions (pure water and seawater). Authors have to explain this and have to provide proof.

Reviewer 2 Report

In general, the article is well written, but a few explanations would be advisable for the readers. I would suggest explaining some doubts:
1. where did the data on the surface roughness of the samples tested start from, was the roughness tested or was it derived from material data? page 3, rows 104
2. specify the source of chemical composition GR15
3. Fig.3. there is no explanation as to what mechanisms of wear dominated with the increase in the frictional distance, for various cases of lubricants. In the case of pure water, the coefficient of friction was initially high, even higher than in the case of dry friction, and then gradually decreased. There is no scientific discussion about this in the text. 

Reviewer 3 Report

The article is written at a high scientific level. Corrections and additions are not required.

Author Response

Dear Professors:

    Thank you very much!

Reviewer 4 Report

The manuscript by Han et al. is dealing with tribological characteristics of composite ceramics important for seawater marine engineering applications. 

The manuscript has lots of testing and analytical explanations of the properties relevant to application. 

Details to be corrected:

indexes for formulas;

references- check what is name and what is surname;

Is there any effect of temperature? Dry and wet friction?

Round 2

Reviewer 1 Report

Authors corrected the manuscript according to reviewers` instruction, but did not take into account the following two comments, and made new formatting mistake: 

--- Scale bars are missing in Figures 5 and 6a. Authors inserted magnification not the scale bar into these figures. Scale bars are more informative. 

--- Authors did not answer the following comment: "Table 5 and text: It is difficult to accept that H3BO3 forms under dry conditions from BN, but B2O3 forms under wet conditions (pure water and seawater). Authors have to explain this and have to provide proof." The question more precisely: Why do you believe that at the presence of large amounts of water, wet conditions, the B2O3 does not react with water to form H3BO3, but when you have only small amounts of water at all, dry conditions, the B2O3 not only forms but reacts in a consecutive step with water to produce H3BO3. Just the opposite is expected. Consider if you can decide on the basis of XPS data if B2O3 or H3BO3, or their mixture forms, and comment it accordingly. I do not agree with you at all that under dry conditions you get the acid and under wet conditions the acid anhydride.  

--- Symbols are still wrong throughout the paper: only the internal quantum number is in subscript, e.g. B 1s, Si 2p, Fe 2p3/2. There is a space between e.g. B and 1s, etc. Please check textbooks. 
